# A Rare Homozygous AP4S1 Variant in Rwandan Siblings with Autosomal Recessive Hereditary Spastic Paraplegia Type 52 (SPG52)

**DOI:** 10.3390/genes16050542

**Published:** 2025-04-30

**Authors:** Sylvine Niyoyita, Esther Uwibambe, Janvier Ndinkabandi, Placide Sesonga, Josse Belladone Niyongere, Benjamin Tuyishimire, Adelaide Urugwiro, Alype Rwamatwara, Gisèle Isingizwe, Janvière Mutamuliza, Christian Nsanzabaganwa, John Bukuru, Florent Rutagarama, Agnès Mukaruziga, Osée Karangwa, Augustin Ndatinya, Maurice Nsanzabera, Norbert Dukuze, Léon Mutesa

**Affiliations:** 1Centre for Human Genetics, School of Medicine and Pharmacy, College of Medicine and Health Sciences, University of Rwanda, Kigali 4285, Rwanda; sylvineniyoyita@gmail.com (S.N.); uwaesther04@gmail.com (E.U.); ndinj12@gmail.com (J.N.); josseniyong@gmail.com (J.B.N.); benjamin.tuyishimire1994@gmail.com (B.T.); adelaiderugwiro1@gmail.com (A.U.); alype.rwamatwara@gmail.com (A.R.); isingisele@gmail.com (G.I.); mamywane@gmail.com (J.M.); nsanzechriss@gmail.com (C.N.); norbertduk123@gmail.com (N.D.); 2Department of Paediatrics and Clinical Genetics, Rwanda Military Referral and Teaching Hospital, Kigali 3377, Rwanda; rutagaramaflorent@gmail.com (F.R.); mukarnesaa@gmail.com (A.M.); augustin.ndatinya@gmail.com (A.N.); nsamaly@yahoo.fr (M.N.); 3Clinical Division, University of Global Health Equity, Kigali 6955, Rwanda; 4Department of Ear, Nose, and Throat, Rwanda Military Referral and Teaching Hospital, Kigali 3377, Rwanda; johnbkrdr@gmail.com

**Keywords:** hereditary spastic paraplegia, SPG52, *AP4S1*, autosomal recessive, genetic testing, Rwanda

## Abstract

**Background/Objectives**: Hereditary spastic paraplegia type 52 (SPG52) is a rare, inherited neurodevelopmental condition passed down in an autosomal recessive pattern. In this report, we describe two siblings from Rwanda who exhibited classic signs of the disorder, including progressive lower-limb spasticity, significant delays in motor development, and exaggerated deep tendon reflexes. **Methods**: Genetic testing through Whole-Exome Sequencing (WES) reveals a rare homozygous splice-site variant (NM_001128126.3:c.295-3C>A) in the *AP4S1* gene. **Results**: Despite the severity of symptoms, both children responded positively to treatment with muscle relaxants and regular physiotherapy. Notably, MRI scans of the brain and spine showed no structural abnormalities. **Conclusions**: By documenting this case, we add to the growing understanding of SPG52, particularly within under-represented Sub-Saharan African populations, and underscore the critical role of early genetic testing in guiding timely diagnosis and intervention.

## 1. Introduction

Hereditary spastic paraplegia type 52 (SPG52) is a highly rare autosomal recessive neurodevelopmental disorder. It begins with neonatal hypotonia, which evolves into hypertonia and spasticity in early childhood. Affected individuals often experience developmental delays, intellectual disabilities, speech difficulties, in some cases mutism, and occasionally febrile seizures [1,2,3]. This condition is part of a group known as adaptor protein complex 4-associated hereditary spastic paraplegias (AP-4 HSP), which includes four subtypes: spastic paraplegia 47 (SPG47), spastic paraplegia 50 (SPG50), spastic paraplegia 51 (SPG51), and spastic paraplegia 52 (SPG52), each caused by single-gene mutations in the Adaptor-Related Protein Complex 4 Subunit β 1 (AP4B1)*, Adaptor-Related Protein Complex 4 Subunit Mu 1(AP4M1)*, Adaptor-Related Protein Complex 4 Subunit Epsilon 1 (AP4E1)*, and Adaptor-Related Protein Complex 4 Subunit Sigma 1 (AP4S1)* genes, respectively [3,4,5,6,7]. A variant in the AP4S1 gene (NM_001128126.3:c.295-3C>A) [8] was identified in African siblings presenting with SPG52. They had features of spastic paraplegia, developmental delay, and intellectual disability [9]. This indicates that there may be a founder effect in African populations, which requires further research to determine the frequency and define its characteristics. The disease presents a wide range of clinical features due to the variety of genes involved, inheritance patterns, and ages of onset [1,2,3,4,5]. The prevalence of SPG52 falls within the broader range of hereditary spastic paraplegias, estimated at 0.1 to 9.6 per 100,000 individuals globally [10,11,12,13]. While its diagnosis is more common in regions with advanced genetic testing, such as Europe and North America, its prevalence remains largely unknown in Sub-Saharan Africa due to diagnostic limitations [1,2,3]. Restricted access to genetic testing in Sub-Saharan Africa hinders diagnosis of hereditary spastic paraplegia. Rwanda has few practicing human geneticists, and testing capabilities are very limited. Only a few hospitals and laboratories have the facilities to conduct tests like karyotyping or Sanger sequencing, but the advanced testing capacity necessary for diagnosing rare diseases such as HSP is not yet present. Advanced genetic tests such as whole-exome sequencing are commissioned to a foreign laboratory, which is both costly and time-consuming. Therefore, the majority of HSP patients might be misdiagnosed or undiagnosed, hence delaying treatment [14].

This report details a case of a rare homozygous variant of the AP-4-associated subtype SPG52 in two siblings of Rwandan descent. To our knowledge, this is the first genetically confirmed SPG52 report from Rwanda and among the few documented in Sub-Saharan Africa.

## 2. Materials and Methods

### 2.1. Subjects and Clinical Assessment

The subjects were enrolled at the Pediatrics and Clinical Genetics Department of the Rwanda Military Referral and Teaching Hospital based on their availability and their legal guardians’ willingness to participate. As the subjects had intellectual disabilities, Informed consent was obtained from the legal guardians. The enrolment was carried out as part of a study investigating the genetic basis of their condition to advance scientific and diagnostic understanding.

Both affected siblings underwent a detailed clinical evaluation in our facility. The assessment comprised history, physical examination, neurological examination, and relevant diagnostic testing.

The neurological exam evaluated the motor system using the Modified Ashworth Scale to assess muscle tone, strength, and spasticity. Deep tendon reflexes were evaluated by striking them with a hammer, and Babinski signs were examined. During the speech examination, dysarthria was noted, and coordination tests were conducted as needed. Developmental milestones were assessed through a caregiver report to identify early cognitive and motor delays. Functional status was measured by observing ambulation, muscle wasting, joint deformities, and performance in daily activities.

The diagnosis of hereditary spastic paraplegia (HSP) was established by clinical presentation according to Harding’s classification, differentiating pure HSP—defined as progressive lower limb spasticity and weakness with no additional neurological features—from complex HSP, including spasticity with intellectual disability, seizures, speech disorder, and limb deformity. Participants in the study fulfilled the criteria for complex autosomal recessive HSP, with evidence of early-onset spastic paraparesis, dysarthria, developmental delay, limb deformity, and epilepsy.

Differential diagnoses of cerebral palsy, leukodystrophies, and spinal cord lesions were considered but ruled out based on the progression of symptoms, lack of perinatal complications, and distinguishing clinical features of spasticity in lower limbs, developmental delay, and seizures. These findings, in conjunction with negative consanguinity but positive family history, favored a clinical diagnosis of complicated HSP.

### 2.2. Imaging and Molecular Investigations

**Imaging Analysis:** Magnetic Resonance Imaging (MRI) scans of the brain and spine were performed to rule out structural abnormalities.

**Sample Collection**: Peripheral blood samples were collected from the siblings at our facility.

**Genomic DNA Extraction and Whole-Exome Sequencing (WES):** Genomic DNA was extracted from each patient’s sample at Centogene’s certified laboratory in Germany. The DNA was then fragmented using enzymatic sonication, and a sequencing library was prepared following standard protocols. Although the specific enrichment kit and Illumina platform used were not disclosed, the testing was performed in accordance with established clinical sequencing guidelines. Sequencing was conducted with an average coverage sufficient to detect variants across targeted regions reliably, and the final libraries were sequenced on an Illumina platform. The precise model (e.g., HiSeq or NextSeq) was not specified in the clinical documentation.

**Ataxia/Spastic Paraplegia Comprehensive Panel:** The Ataxia/Spastic Paraplegia Panel included the coding regions of selected genes, ±10 base pairs of flanking intronic sequences, and known pathogenic/likely pathogenic non-coding variants. The gene panel encompassed *Atrophin 1 (ATN1), Ataxin 1 (ATXN1), Ataxin 10 (ATXN10), Ataxin 2 (ATXN2), Ataxin 3 (ATXN3), Ataxin 7 (ATXN7), Ataxin 8 Opposite Strand (ATXN8OS), Brain-Expressed, Associated with NEDD4, 1 (BEAN1), Calcium Voltage-Gated Channel Subunit Alpha1 A (CACNAIA), Frataxin (FXN), NOP56 Ribonucleoprotein (NOP56), Protein Phosphatase 2 Regulatory Subunit B β (PPP2R2B),* and the *TATA-Box-Binding Protein (TBP) gene*, among others. The Ataxia/Spastic Paraplegia Comprehensive Panel was provided by Centogene, which includes sequencing and analysis of the genes associated with hereditary ataxia and spastic paraplegia [15]. The panel includes genes associated with both autosomal dominant and autosomal recessive forms of hereditary spastic paraplegia (HSP) and cerebellar ataxia. The complete list of genes analyzed is provided in Table 1.

This targeted approach not only confirmed the presence of a rare homozygous variant in the AP4S1 gene (NM_001128126.3:c.295-3C>A) but also effectively ruled out other possible genetic causes with overlapping phenotypes.

Repeat expansion analysis was conducted using PCR and capillary electrophoresis for key genes (*ATXN1*, *ATXN2*, *FXN*, etc.), and a repeat-primed assay (RPA) was performed to screen for large pathogenic expansions where applicable.

**Variant Calling and Annotation Pipeline:** Sequencing reads were mapped to the human reference genome (GRCh37/hg19) and mitochondrial genome. Variants were called and annotated using the Centocloud Bioinformatics pipeline, Centogene’s validated in-house software, and classified based on the 2015 ACMG guidelines.

**Copy Number and Mitochondrial Variant Detection:** Copy number variants (CNVs) were detected using a proprietary CNV-calling algorithm validated by Centogene, with >95% sensitivity for homozygous/hemizygous and heterozygous events spanning at least three consecutive exons. Mitochondrial variants were reported for heteroplasmy levels ≥ 15%. Although the exact software was not specified in the report, Centogene’s CNV detection pipeline meets clinical diagnostic sensitivity standards.

**Splice Effect Prediction:** The splice site effect was analyzed in terms of Ada and RF scores. The variant (NM_001128126.3:c.295-3C>A) was a ‘2/2 likely splice effect’, indicating that both programs agree it impacts splicing. The variant’s location in a conserved splice acceptor site and its presumed pathogenic classification in ClinVar (Variation ID: 210218) support this interpretation. Regrettably, the clinical report did not include complete parameters for the Ada/RF predictions, nor was the assessment of the splice effect feasible, as a methodological limitation was reported.

**Quality Control Measures:** Variants with low sequencing depth, poor mapping quality, or unclear zygosity were confirmed by orthogonal methods. However, no specific orthogonal method was reported by the lab facility. Specificity for all reported variants exceeded 99.9%, as guaranteed by the laboratory’s internal validation protocols.

## 3. Results

### 3.1. Clinical Assessments

Detailed personal histories, including familial pedigree (see Figure 1) and medical records, were obtained from the study’s participants. Physical examinations, including neurological examination, were conducted to assess neurological signs, including muscle tone, strength, and spasticity following the Modified Ashworth Scale, and deep tendon reflexes and Babinski responses were assessed.

*First sibling*: A 28-year-old female presented with long-standing progressive spastic paraparesis from 5 years of age as the age of onset, followed by paraplegia consistent with hereditary spastic paraplegia. Her family history was unremarkable for similar symptoms, and there was no parental consanguinity. (However, the claim that the family history is unremarkable and that there is no consanguinity might be affected by cultural biases, especially considering the potential under-reporting or misinterpretation of consanguinity in certain populations. Both parents originate from rural Rwanda, where social perceptions or limited genetic literacy may influence how consanguineous relationships are disclosed or understood.) Early developmental delays were noted during childhood. Physical examination at first presentation revealed slurred speech, significant spasticity in the lower limbs with non-pitting edema and deformity, heightened deep tendon reflexes, and extensor plantar responses. She was wheelchair-dependent.

*Second sibling*: A 26-year-old male presented with long-standing progressive spastic paraparesis from 3 years of age as the age of onset, and later, paraplegia consistent with hereditary spastic paraplegia. His family history was also unremarkable for similar symptoms. His early developmental delays were noted during childhood. He presented similar clinical features to his older sister. Physical examination at first presentation revealed a flaccid individual with slurred speech, significant spasticity in the lower limbs associated with deformity, heightened deep tendon reflexes, and extensor plantar responses. He was also wheelchair-dependent.

The two siblings (Figure 2) were treated with the anti-epileptic drug phenobarbital to manage seizures. The findings point to progressive upper motor neuron signs predominantly in the lower limbs without a clear radiological or biological cause, suggesting features like hereditary spastic paraplegia (SPG). Genetic testing was initiated despite a negative family history to investigate potential genetic contributions.

### 3.2. Imaging Analysis

MRI scans of the brain and spine for both siblings did not reveal any abnormalities.

### 3.3. Exome Sequencing Analysis

Sequence analysis reveals a rare homozygous variant in the AP4S1 gene (NM_001128126.3:c.295-3C>A), located just upstream of the canonical splice site. This variant is predicted to affect splicing at the affected exon 4, though the exact protein-level impact remains unclear.

### 3.4. Pathogenicity Interpretation

The variant **NM_001128126.3:c.295-3C>A** is classified as *likely pathogenic* by the Ada/RF scores and supported by conservation analysis with a score of 2/2, suggesting that the affected site is highly conserved and that the variant may disrupt normal splicing. However, the tool used by the lab facility for conservation analysis is not specified.

Population frequency data from the **Genome Aggregation Database (gnomAD)** indicate that this variant is extremely rare, with a global allele frequency of **0.000012**. However, in the **African population**, its frequency is **0.0001734 [16]**. To further support the clinical interpretation, the **CentoMD^®^** proprietary database was consulted, offering curated genetic and clinical data. The pathogenicity of this variant is also evidenced by its previous detection in affected African siblings, in which functional analysis revealed that it resulted in aberrant canonical AP4S1 splicing, resulting in the loss of isoform 2 [9]. The phenotypic overlap with the current patients, namely, spasticity, developmental delay, and wheelchair dependence, lends support to its causative role in SPG52. Following the **ACMG/AMP/ClinGen SVI variant classification guidelines**, the variant was interpreted as a likely pathogenic *splicing-type* mutation (Class 2), as summarized in Table 2.

The PS3 classification is supported by functional predictions indicating altered splicing, while PM2 is met due to the variant’s absence or extremely low frequency in gnomAD. PP3 is met by the conservation scores and two splice prediction tools. PP5 reflects its inclusion in CentoMD^®^ as a likely pathogenic variant [17].

## 4. Discussion

Hereditary spastic paraplegia type 52 (SPG52) is an autosomal recessive neurodevelopmental disorder caused by mutations in the *AP4S1* gene, which encodes a subunit of the adaptor protein complex 4 (AP-4) [1,2,18]. In this study, we report a rare homozygous variant (NM_001128126.3:c.295-3C>A) in the *AP4S1* gene in two patients, who are siblings of Rwandan descent, with hereditary spastic paraplegia (SPG). The observed phenotype included early-onset progressive lower limb spasticity, developmental delay, hyper-reflexia, and gait abnormalities, clinical features that are consistent with autosomal recessive hereditary spastic paraplegia type 52 (SPG52), previously associated with *AP4S1* variants [1,2]. Of specific interest, the identical AP4S1 variant (c.295-3C>A) was described in two African sisters with SPG52, who manifested with progressive spastic paraplegia, profound developmental delay, and foot deformities, but with an earlier age of onset (4–5 years) than our patients (3–5 years) [9]. The two groups had several features in common, such as wheelchair dependence, intellectual disability, and lack of structural MRI abnormalities, suggesting a uniform core phenotype. However, variability in the expressivity of the condition is highlighted by differences in seizure occurrence (present in our cases but not in the preceding report) and speech difficulty (dysarthria rather than mutism). The recurrence of the variant in Africans (gnomAD African allele frequency: 0.0001734; absent in non-Africans) [16] is suggestive of a local founder effect, with a requirement for large-scale investigations in Sub-Saharan Africa.

To contextualize these observations, we compared the clinical features of our Rwandan siblings with those reported in European SPG52 cohorts (Table 3).

Of specific interest, our patients had an older age of onset (3–5 years) compared to the European cases (typically 6 months–3 years), although both cohorts had severe progression and wheelchair dependence. This could be a result of genetic modifiers, environmental factors, or a delay in diagnosis in resource-limited settings.

Whole-exome sequencing (WES) identified this intronic variant near the splice acceptor site of exon 4. The Ada and RF scores indicate disruption of a splice site (2/2). This aligns with the conserved location of the variant as a splice site and ClinVar classification (Variation ID: 210218), although the scoring parameters are not available. As no amino acid change was detected, further investigation through RNA sequencing (RNA-Seq) for gene expression and splicing analysis, functional assays, and family studies is recommended. This underscores the importance of genetic testing in hereditary spastic paraplegia type 52 to address diagnostic challenges arising from its genetic complexity and overlapping clinical features.

The clinical features observed in our two Rwandan siblings, including early-onset spasticity, motor developmental delay, intellectual disability, and seizures, are consistent with previous reports of SPG52 caused by *AP4S1* mutations. Seizures, while not universally present in all cases, have been frequently reported in association with AP-4 complex deficiencies, including SPG52, further supporting the pathogenicity of the identified NM_001128126.3:c.295-3C>A variant [15,19]. Both patients in our study are currently managed with phenobarbital, highlighting the relevance of seizure management in this condition. These findings not only align with the broader phenotypic spectrum described in *AP4S1*-related disorders but also underscore the importance of comprehensive neurological assessment in suspected HSP cases.

Genetic testing is essential for accurately diagnosing spastic paraplegia, as it is genetically diverse and caused by several gene mutations [2,3]. Therefore, the presence of homozygous pathogenic mutations in the *AP4S1* gene has confirmed the diagnosis of spastic paraplegia, aligning with the patient’s clinical presentation. Studies indicated that homozygous pathogenic mutations in the *AP4S1* gene are associated with neonatal hypotonia that progresses to hypertonia and spasticity in early childhood, developmental delay, mental retardation, and poor or absent speech. Febrile or afebrile seizures may also occur, collectively well-known signs and symptoms of spastic paraplegia [1,2,3,4].

The confirmation of autosomal recessive spastic paraplegia type 52 (SPG52) through combined molecular genetic tests and comprehensive clinical examinations underscores the role of the *AP4S1* gene, which encodes the small subunit of the adaptor protein complex 4 (AP4 complex). AP-4 is part of a conserved family of heterotetrameric protein complexes (AP-1 to AP-5) involved in the selective incorporation of transmembrane cargo proteins into vesicles and their trafficking within cells. Comprising subunits β4, ε, μ4, and σ4, AP-4 functions at the trans-Golgi network to facilitate vesicle trafficking to endosomes or the basolateral plasma membrane independently of clathrin [5,6]. Disruptions in any of the complex’s subunits, including *AP4S1*, are associated with similar autosomal recessive phenotypes primarily characterized by spastic tetraplegia [2,3,4,5,6,7,13]. Despite the absence of pertinent familial history, the development history was inclusive to differentiate from certain subtypes of hereditary spastic paraplegia [2,3,4,5,6,7].

Although seizures and speech impairment are present in our patients, they may be absent in others [5]. This highlights the phenotypic variability of *AP4S1*-related disorders and underscores the importance of individualized clinical assessment. The pair demonstrated significant improvement with muscle relaxants, physical therapy, and exercises, underscoring the criticality of timely identification and intervention in the management of spastic paraplegia. Spastic paraplegia patients can greatly improve their quality of life by showing promising results with muscle relaxants and intensive physical therapy [14,20,21]. Although no cure has been identified at the moment, supporting management (*by a multidisciplinary team, including a neurologist, a clinical geneticist, a developmental specialist, an orthopedic surgeon/physiatrist, a physical therapist, an occupational therapist, and a speech and language pathologist*), such as physiotherapy and the use of muscle relaxants, can help alleviate the effects encountered in the condition [5,6,7,20].

As no amino acid change was detected, further testing for non-coding variants, splice site variants, copy number variants, and epigenetic changes is recommended. Performing RNA sequencing (RNA-Seq) for gene expression and alternative splicing analysis and conducting functional assays to assess protein function are essential. Additionally, exploring family studies for co-segregating variants and evaluating non-genetic factors that may contribute to the condition.

Although the parents of the siblings are not consanguineous, they originate from the same rural region of Rwanda, raising the possibility of shared ancestry and a potential regional founder effect. Its absence in non-African gnomAD populations (global frequency: 0.000012; African: 0.0001734) and recurrence in these siblings further support this hypothesis. Regional studies are needed to assess its prevalence in Rwandan subpopulations [16]. Large-scale population studies across Rwanda and Sub-Saharan Africa are needed to determine the frequency of this variant in ethnically and geographically matched controls. Such data would greatly aid in variant interpretation and enhance genetic counseling in African populations, which remain significantly under-represented in global genetic databases [14].

This case report adds to the growing body of literature by documenting a rare *AP4S1* variant in non-consanguineous siblings of Rwandan descent with hereditary spastic paraplegia type 52. It highlights diagnostic challenges posed by the condition’s genetic diversity and overlapping clinical features while emphasizing the importance of molecular testing in achieving an accurate diagnosis and avoiding misclassification. Given the rarity of this variant and its occurrence in African individuals, the findings underscore the critical need for population-specific reference databases and studies that explore founder mutations in under-represented regions such as Sub-Saharan Africa.

## 5. Conclusions

This case report focuses on the clinical, diagnostic, and management aspects of spastic paraplegia type 52 linked to a rare variant homozygous variant mutation (NM_001128126.3:c.295-3C>A) in the *AP4S1* gene in two siblings from Rwanda. The re-emergence of the c.295-3C>A variant in African patients with SPG52 emphasizes the necessity of population-specific genetic databases and founder effect research in under-represented regions. Collaborative research to standardize such variants in Sub-Saharan Africa will improve diagnostic sensitivity and genetic counseling [20,21]. Given the genetic diversity of the condition and its potential to stem from various gene mutations, our findings add valuable insights to the existing knowledge, especially within the African population.

## Figures and Tables

**Figure 1 genes-16-00542-f001:**
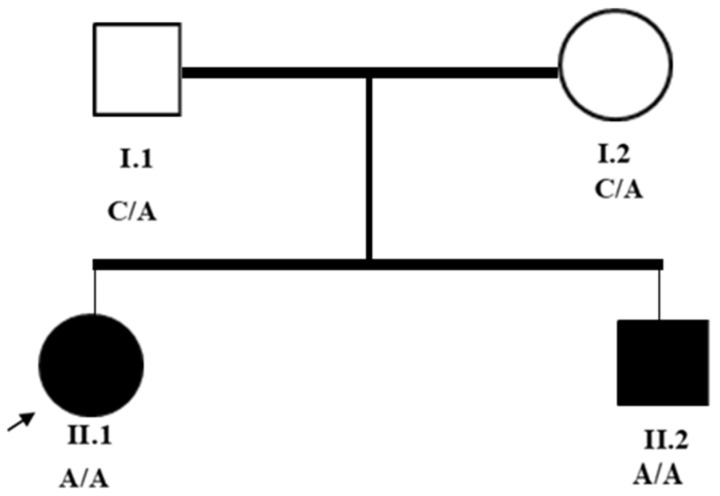
Pedigree illustrates an autosomal recessive pattern. Squares are males, circles are females; solid symbols are affected individuals. Roman numbers are utilized to designate generations (I and II), and Arabic numbers (1 and 2) designate individuals. C (cytosine) and A (adenine) are the nucleotide bases involved in the mutation.

**Figure 2 genes-16-00542-f002:**
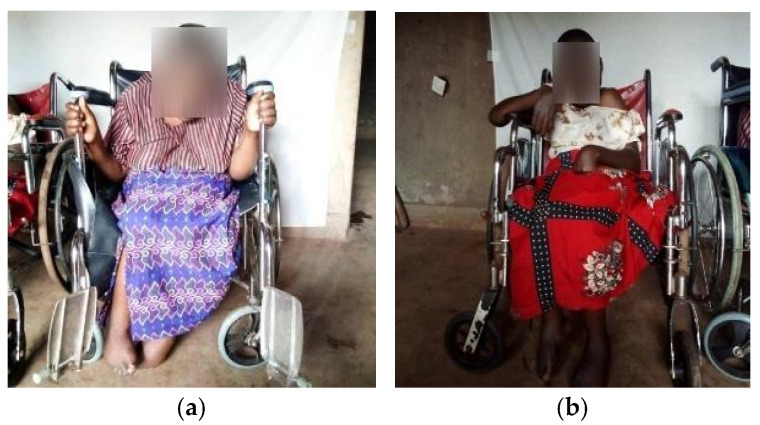
Cases’ presentation: (**a**) First sibling presenting limb deformities, limited movement, and wheelchair-dependent situation; (**b**) Second sibling with a similar clinical presentation as the first sibling.

**Table 1 genes-16-00542-t001:** List of known genes implicated in hereditary spastic paraplegia and cerebellar ataxia, including their associated disorders and inheritance patterns.

Gene Symbol	Gene Name	Associated Disorder(s)	Inheritance Mode
ATN1	Atrophin 1	Dentatorubral–Pallidoluysian Atrophy (DRPLA)	Autosomal Dominant
ATXN1	Ataxin 1	Spinocerebellar Ataxia Type 1 (SCA1)	Autosomal Dominant
ATXN2	Ataxin 2	Spinocerebellar Ataxia Type 2 (SCA2), ALS Modifier	Autosomal Dominant
ATXN3	Ataxin 3	Spinocerebellar Ataxia Type 3 (SCA3, Machado–Joseph Disease)	Autosomal Dominant
ATXN7	Ataxin 7	Spinocerebellar Ataxia Type 7 (SCA7)	Autosomal Dominant
ATXN8OS	Ataxin 8 Opposite Strand	Spinocerebellar Ataxia Type 8 (SCA8)	Autosomal Dominant (Incomplete Penetrance)
ATXN10	Ataxin 10	Spinocerebellar Ataxia Type 10 (SCA10)	Autosomal Dominant
BEAN1	Brain-Expressed and Associated with NEDD4, 1	Spastic Paraplegia 75	Autosomal Recessive
CACNA1A	Calcium Voltage-Gated Channel Subunit Alpha1 A	Episodic Ataxia, SCA6, Familial Hemiplegic Migraine	Autosomal Dominant
FXN	Frataxin	Friedreich Ataxia	Autosomal Recessive
NOP56	NOP56 Ribonucleoprotein	Spinocerebellar Ataxia Type 36 (SCA36)	Autosomal Dominant
PPP2R2B	Protein Phosphatase 2 Regulatory Subunit B β	Spinocerebellar Ataxia Type 12 (SCA12)	Autosomal Dominant
TBP	TATA-Box-Binding Protein	Spinocerebellar Ataxia Type 17 (SCA17)	Autosomal Dominant
AP4S1	Adaptor-Related Protein Complex 4 Subunit Sigma 1	Spastic Paraplegia 52 (SPG52)	Autosomal Recessive

**Table 2 genes-16-00542-t002:** Variant classification based on ACMG/AMP/ClinGen SVI guidelines.

Criterion Code	Description	Strength *
Pathogenic Evidence		
PS3	Functional studies are supportive of a damaging effect	Strong
PM2	Absent or very low frequency in large population databases	Moderate
PP3	Multiple lines of computational evidence support a deleterious effect	Supporting
PP5	A variant reported in reputable sources as pathogenic	Supporting

* Interpretation rules: Likely pathogenic: 1—strong (PS), 1–2—moderate (PM), or supporting (PP).

**Table 3 genes-16-00542-t003:** Clinical comparison of SPG52 shows later onset in Rwandan siblings (3–5 years) versus European cases (6 months–3 years), with similarly severe progression and comparable treatment response (Tessa et al., 2021; Abou Jamra et al., 2011) [5].

Feature	European Cases (Tessa et al., 2021; Abou Jamra et al., 2011) [5]	Rwandan Siblings (This Study)
**Age of Onset**	6 months–3 years (neonatal hypotonia → childhood spasticity)	3–5 years (delayed motor milestones)
**Severity**	Moderate–severe (wheelchair dependence common)	Severe (wheelchair-dependent by adulthood)
**Seizures**	Reported in ~50% of cases	Present (managed with phenobarbital)
**Speech Delay**	Frequent (mutism in some)	Present (dysarthria)
		Similar response to therapy
**Treatment Response**	Partial improvement with physiotherapy/muscle relaxants	

## Data Availability

The original contributions presented in this study are included in the article. Further inquiries can be directed to the corresponding author.

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
