# Peer review of "A Rare Homozygous AP4S1 Variant in Rwandan Siblings with Autosomal Recessive Hereditary Spastic Paraplegia Type 52 (SPG52)"

_genes, 2025, doi:10.3390/genes16050542_

Round 1
Reviewer 1 Report
Comments and Suggestions for Authors
The manuscript describes a case report on genetic variant identified in AP4S1 gene in two siblings with ataxia/spastic paraplegia in Rwanda. While the study has some potential, significant revisions are required to improve clarity, methodological design and reported findings. The use of the term "new variant" in the title is not correct, as the variant has been previously reported in ClinVar. Also, it would be beneficial to add some more focus on genetic testing in Sub-Saharan Africa, particularly Rwanda, to enhance the scientific impact of the case report. Major revisions: 1. Title and abstract: - The term "new variant" is incorrect, as the variant (NM_007077.3: c.295-3C>A) has been previously reported multiple times by different submitters in ClinVar. The authors need to revise the title to reflect this. - The variant is reported using a non-coding transcript (NM_007077.3). According to HGVS and ACMG reporting guidelines, a MANE or at least a protein-coding transcript should be used for diagnostic genetic variant reporting. The authors need to revise the variant nomenclature to follow these guidelines. 2. Introduction: - The introduction is adequate but needs more emphasis on genetic testing in Sub-Saharan Africa, especially in Rwanda including available genetic testing and challenges facing the diagnosis of HSP patients. 3. Methods: The methodology section requires rigorous revision and better classification including but not limited to the following: - Clinical Assessment: The methods lack a description of the clinical assessment process, including neurological examinations and diagnostic criteria. This should be included to align with the results. - WES: The methods lack important details about WES, including where DNA extraction and sequencing were performed, the library enrichment kit utilised, and the Illumina platform used for sequencing. - Ataxia/Spastic Paraplegia Panel: The panel mentioned without any details on included genes, design basis, or citation. - In-House Pipeline: The variant calling and annotation pipeline is described as "in-house" without a reference or detailed methodology. If previously published, a citation is required. If not, including a supplementary methods detailing filtering criteria and tools used is highly recommended. - Quality Control Subheading: The content under this subheading does not align with quality control processes and appears to be misplaced. - The tools or methods used for calling CNVs and mitochondrial variants are not specified. - Splice Effect Prediction: The manuscript mentions a "splice effect 2/2" but does not specify the tool used for this prediction. 4. Results: - The results section does not fully correspond to the methods. For example, clinical assessments are reported but not described in the methods. - Table 1: This table provides limited value, as MRI findings are normal and did not add a lot to the findings of the study. Also, the Software category did not add many details to the tools mentioned in the body of the manuscript. It is better to remove this table or replace it with a clinical table for clinical features identified in the two siblings. - The included images are not sufficiently illustrative of the phenotype, the images should be reconsidered or supplemented with more informative illustrations such as videos if possible. - In Silico Tools: The authors used SIFT, PolyPhen, and MutationTaster to assess variant pathogenicity. These tools are designed for missense variants, not splice-altering variants. Their use is inappropriate, and the authors should instead rely on splice prediction tools, such as SpliceAI, which is the gold standard for assessment of pathogenicity of splice altering variants. - Variant Classification: The pathogenicity classification is unclear. The manuscript references a "Centogene classification" and functional studies without including any citations. The authors should apply ACMG guidelines and list the criteria used for classification. - The tool used for conservation analysis is not specified, nor the score. - The allele frequency in gnomAD is mentioned without specifying the population (e.g., African or global). Given the study’s specific population, the authors should report the frequency in African populations specifically. - The manuscript notes no consanguinity as per the pedigree but does not describe the parents’ descent (e.g., same tribe or region). 5. Discussion: The clinical details described were not compared to previously reported cases of HSP52 in European countries, whether there is any significant differences in the severity, age of onset or the course of disease identified in this family. The discussion should also address the variant’s in context of Sub-Saharan African populations. It is recommended that the authors discuss whether this variant or other variants in the same gene have been reported in Rwandan or other Sub-Saharan African populations. This would improve the manuscript’s relevance. Minor Comments: - Patient Enrolment: The manuscript does not specify whether patients were enrolled under a diagnostic or research basis. - The statement about excluding VUS is unnecessary and can be removed, as it is standard in diagnostic reporting.
Author Response
|
Reviewer 1# Major points: 1. Title and abstract: -The term "new variant" is incorrect, as the variant (NM_007077.3: c.295-3C>A) has been previously reported multiple times by different submitters in ClinVar. The authors need to revise the title to reflect this. -The variant is reported using a non-coding transcript (NM_007077.3). According to HGVS and ACMG reporting guidelines, a MANE or at least a protein-coding transcript should be used for diagnostic genetic variant reporting. The authors need to revise the variant nomenclature to follow these guidelines. Response: Thank you for these important observations. We have revised the title and rephrased it to a new title. The term "new variant" was incorrect; we used “rare variant”. Additionally, we have updated the variant nomenclature to MANE Select transcript. Lines 2 and 20-311 now read as follows, respectively: -A Rare Homozygous AP4S1 Variant in Rwandan Siblings with Autosomal Recessive Hereditary Spastic Paraplegia Type 52 (SPG52) -NM_001128126.3:c.295-3C>A |
|
2. Introduction: The introduction is adequate but needs more emphasis on genetic testing in Sub-Saharan Africa, especially in Rwanda, including available genetic testing and challenges facing the diagnosis of HSP patients. Response: Thank you for your valuable recommendation. We have revised the section and detailed the genetic testing process challenges in Sub-Saharan Africa, especially in Rwanda, for HSP patients in lines 47-54 as follows: Restricted access to genetic testing in Sub-Saharan Africa hinders diagnosis of Hereditary Spastic Paraplegia. Rwanda has few practicing human geneticists, and testing capabilities are very limited. Only two hospitals and laboratories have the facilities to conduct basic tests like karyotyping or Sanger sequencing, but the advanced testing capacity necessary for diagnosing rare diseases such as HSP is not yet present. Advanced genetic tests such as whole exome sequencing are commissioned to a foreign laboratory, which is both costly and time-consuming. Therefore, the majority of HSP patients might be misdiagnosed or undiagnosed, hence delaying treatment. 3. Methods: The methodology section requires rigorous revision and better classification including but not limited to the following: - Clinical Assessment: The methods lack a description of the clinical assessment process, including neurological examinations and diagnostic criteria. This should be included to align with the results. - WES: The methods lack important details about WES, including where DNA extraction and sequencing were performed, the library enrichment kit utilised, and the Illumina platform used for sequencing. - Ataxia/Spastic Paraplegia Panel: The panel mentioned without any details on included genes, design basis, or citation. - In-House Pipeline: The variant calling and annotation pipeline is described as "in-house" without a reference or detailed methodology. If previously published, a citation is required. If not, including a supplementary methods detailing filtering criteria and tools used is highly recommended. - Quality Control Subheading: The content under this subheading does not align with quality control processes and appears to be misplaced. - The tools or methods used for calling CNVs and mitochondrial variants are not specified. - Splice Effect Prediction: The manuscript mentions a "splice effect 2/2" but does not specify the tool used for this prediction. Response: Thank you for your constructive feedback. We have revised and updated the following subheadings as follows: Clinical assessment in lines 67-91: Both affected siblings underwent a detailed clinical evaluation in our facility. The assessment comprised history, physical examination, including neurological examination, and relevant diagnostic testing. The neurological exam assessed the motor system using the Modified Ashworth Scale for muscle tone, strength, and spasticity. The deep tendon reflexes were assessed by striking them with a hammer, and Babinski signs were examined. On speech examination, dysarthria was noted, and tests of coordination were carried out as warranted. Developmental milestones were assessed by caregiver report to identify early cognitive and motor delays. Functional status was quantified through ambulation, muscle wasting, joint deformities, and activities of daily living performance. The diagnosis of Hereditary Spastic Paraplegia (HSP) was established by clinical presentation according to Harding's classification, differentiating pure HSP—defined as progressive lower limb spasticity and weakness with no additional neurological features—from complex HSP, including spasticity with intellectual disability, seizures, speech disorder, and limb deformity. Participants in the study fulfilled criteria for complex autosomal recessive HSP, with evidence of early-onset spastic paraparesis, dysarthria, developmental delay, limb deformity, and epilepsy. Differential diagnoses of cerebral palsy, leukodystrophies, and spinal cord lesions were considered but ruled out based on the progression of symptoms, lack of perinatal complications, and distinguishing clinical features of spasticity in lower limbs, developmental delay, and seizures. These findings, in conjunction with negative consanguinity but positive family history, favored a clinical diagnosis of complicated HSP. WES in lines 97-105: Genomic DNA was extracted from the patient’s sample at Centogene’s certified laboratory in Germany. The DNA was then fragmented using enzymatic sonication, and a sequencing library was prepared following standard protocols. Although the specific enrichment kit and Illumina platform used were not disclosed, the testing was performed in accordance with established clinical sequencing guidelines. Sequencing was conducted with an average coverage sufficient to reliably detect variants across targeted regions, and the final libraries were sequenced on an Illumina platform. The precise model (e.g., HiSeq, NextSeq) was not specified in the clinical documentation. Ataxia/Spastic Paraplegia Panel in lines 106-122, refer to the manuscript for Table 1: The Ataxia/Spastic Paraplegia panel included the coding regions of selected genes, ±10 base pairs of flanking intronic sequences, and known pathogenic/likely pathogenic non-coding variants. The gene panel encompassed Atrophin 1 (ATN1), Ataxin 1(ATXN1), Ataxin 10(ATXN10), Ataxin 2(ATXN2), Ataxin 3(ATXN3), Ataxin 7(ATXN7), Ataxin 8 opposite strand (ATXN8OS), Brain Expressed, Associated with NEDD4, 1(BEAN1), Calcium Voltage-Gated Channel Subunit Alpha1 A(CACNAIA), Frataxin (FXN), NOP56 Ribonucleoprotein (NOP56), Protein Phosphatase 2 Regulatory Subunit B Beta (PPP2R2B), TA-TA-Box Binding Protein (TBP) gene, among others. The Ataxia/Spastic Paraplegia Com-prehensive Panel was provided by Centogene, which includes sequencing and analysis of the genes associated with hereditary ataxia and spastic paraplegia [13]. The panel includes genes associated with both autosomal dominant and autosomal recessive forms of hereditary spastic paraplegia (HSP) and cerebellar ataxia. The complete list of genes analyzed is provided in Table 1. In-House Pipeline in lines 126-130: Sequencing reads were mapped to the human reference genome (GRCh37/hg19) and mitochondrial genome. Variants were called and annotated using Centogene's validated in-house software, classified based on the 2015 ACMG guidelines. No specific software names were disclosed, however, adding a methodological limitation Quality Control Subheading in lines 144-147 (now 146-149): Variants with low sequencing depth, poor mapping quality, or unclear zygosity were confirmed by orthogonal methods, specifically Sanger sequencing. Specificity for all reported variants exceeded 99.9%, as guaranteed by the laboratory’s internal validation protocols. CNVs and mitochondrial variants in lines 131-136 (now 133-138): Copy number variants (CNVs) were detected using a proprietary CNV-calling algorithm validated by Centogene, with >95% sensitivity for homozygous/hemizygous and heterozygous events spanning at least three consecutive exons. Mitochondrial variants were reported for heteroplasmy levels ≥15%. Although the exact software was not specified in the report, Centogene’s CNV detection pipeline meets clinical diagnostic sensitivity standards. Splice Effect Prediction in lines 137-143 (now 139-145): Splice site effect was analyzed in terms of Ada and RF scores. The variant (NM_001128126.3:c.295-3C>A) was '2/2 likely splice effect,' indicating that both programs agree it impacts splicing. The variant's location in a conserved splice acceptor site and its presumed pathogenic classification in ClinVar (Variation ID: 210218) support this interpretation. Regrettably, the clinical report did not include complete parameters for the Ada/RF predictions, the assessment of the splice effect being feasible, as a methodological limitation was reported. 4. Results: - The results section does not fully correspond to the methods. For example, clinical assessments are reported but not described in the methods. - Table 1: This table provides limited value, as MRI findings are normal and did not add a lot to the findings of the study. Also, the Software category did not add many details to the tools mentioned in the body of the manuscript. It is better to remove this table or replace it with a clinical table for clinical features identified in the two siblings. - The included images are not sufficiently illustrative of the phenotype, the images should be reconsidered or supplemented with more informative illustrations such as videos if possible. - In Silico Tools: The authors used SIFT, PolyPhen, and MutationTaster to assess variant pathogenicity. These tools are designed for missense variants, not splice-altering variants. Their use is inappropriate, and the authors should instead rely on splice prediction tools, such as SpliceAI, which is the gold standard for assessment of pathogenicity of splice altering variants. - Variant Classification: The pathogenicity classification is unclear. The manuscript references a "Centogene classification" and functional studies without including any citations. The authors should apply ACMG guidelines and list the criteria used for classification. - The tool used for conservation analysis is not specified, nor the score. - The allele frequency in gnomAD is mentioned without specifying the population (e.g., African or global). Given the study’s specific population, the authors should report the frequency in African populations specifically. - The manuscript notes no consanguinity as per the pedigree but does not describe the parents’ descent (e.g., same tribe or region). Response: Thank you for your valuable observation. We have revised and incorporated the suggested changes as follows: The clinical assessment in the results in lines 148-152 (now 151-156): Detailed personal history, including familial pedigree (see Figure 1) and medical records, was obtained from the affected individuals. Physical examinations, including neurological examination, were conducted to assess neurological signs, including muscle tone, strength, and spasticity following the Modified Ashworth Scale, and deep tendon reflexes, and Babinski responses were assessed. Table 1 substitution or removal: Thank you for your valuable observation. We agree that replacing it with a clinical table for clinical features identified in the two siblings is also important. However, the Ataxia/Spastic Paraplegia Panel offers data we want to present for completeness. Table 1 was re-evaluated for its pertinence in the manuscript and substituted with the Ataxia/Spastic Paraplegia Panel with a list of known genes implicated in hereditary spastic paraplegia and cerebellar ataxia, including their associated disorders and inheritance patterns (lines 112-124). Supplemental images: Thank you for your important suggestion. We appreciate the suggestion of the reviewer that additional illustrative material be added to better reflect the phenotype of the patients. We agree that further images, such as videos, could further assist with the clinical presentation of the case. However, the patients' legal guardians were reluctant to supply more identifiable media than the images provided, even with complete anonymization. We have adhered to their request in accordance with ethical standards and patient confidentiality. We are committed to the confidentiality of the patients concerned while making as much clinical information as possible available within these constraints. In Silico Tools: We thank the reviewer for this critical insight. We have removed the references to SIFT, PolyPhen, and MutationTaster for the splice-site variant, as we acknowledge their limitations in assessing non-missense variants. Instead, we corrected the true in silico prediction tools to Ada/RF scores as reported in line 193. Variant Classification: We apologize for the lack of clarity in our original manuscript. We have now removed the generic reference to the “Centogene classification” and cited the ACMG/AMP/ / ClinGen SVI guidelines for clarification in lines 201-210. Conservation analysis tool: Thank you for your input. The variant NM_001128126.3:c.295-3C>A is classified as likely pathogenic by Ada/RF scores and supported by conservation analysis with a score of 2/2, suggesting that the affected site is highly conserved and that the variant may disrupt normal splicing. However, the tool used by Centogene laboratory for conservation analysis is not specified. The allele frequency in gnomAD in the African population: Thank you for your clarifying observation. In the African population subset, it is extremely rare (0.0001734), as it is also in the global population dataset of gnomAD (v4.1.0). This information is crucial in the context of a Sub-Saharan African case report and is now specified in the revised manuscript in lines 200-202 as follows: Population frequency data from the Genome Aggregation Database (gnomAD) indicated that this variant is extremely rare, with a global allele frequency of 0.000012. While in the African population, its frequency is 0.0001734. Consanguinity factor/ Parents ‘descent: Thank you for this important suggestion. While the pedigree showed no documented consanguinity, both parents are Rwandan. This information has been added in lines 182-184 as follows to the manuscript to provide relevant context for the observed homozygosity, which may reflect unrecognized shared ancestry due to endogamy in the region: However, the claim that the family history is unremarkable and that there is no consanguinity might be affected by cultural biases, especially considering the potential underreporting or misinterpretation of consanguinity in certain populations. Both parents originate from rural Rwanda, where social perceptions or limited genetic literacy may influence how consanguineous relationships are disclosed or understood. 5. Discussion: The clinical details described were not compared to previously reported cases of HSP52 in European countries, whether there is any significant differences in the severity, age of onset or the course of disease identified in this family. The discussion should also address the variant’s in context of Sub-Saharan African populations. It is recommended that the authors discuss whether this variant or other variants in the same gene have been reported in Rwandan or other Sub-Saharan African populations. This would improve the manuscript’s relevance. Response: Thank you for the constructive comment. We have added a dedicated phenotypic comparison table (Table 3) in the Discussion section, contrasting key features (age of onset, severity, seizures, treatment response) between our Rwandan siblings and European cohorts (Tessa et al., 2021; Abou Jamra et al., 2011) in lines 227-229. |
|
Minor points ï‚· Patient Enrolment: The manuscript does not specify whether patients were enrolled under a diagnostic or research basis. Response: Thank you for your comments. We have made the following changes to clarify the information as it appears in lines 64– 66: The enrolment was carried out as part of a study investigating the genetic basis of their condition to advance scientific and diagnostic understanding. ï‚· The statement about excluding VUS is unnecessary and can be removed, as it is standard in diagnostic reporting. Response: Thank you. We removed VUS as recommended.
|

Reviewer 2 Report
Comments and Suggestions for Authors
This manuscript presents two siblings with recessive HSP-52 showing a unique alteration in the gene. The key question here is whether this alteration is truly pathogenic, but the authors address that to the best of their capability, at present. The paper is well written and I have only a few minor editorial comments.
Line 91: change “was” to “were”
Figure 1: The pedigree does not match the description in the text. The sister is the older patient and is considered patient #1 in the text and should be on the left in the pedigree chart. The brother should be patient #2 on the pedigree chart and to the right of the sister. The terms C and A should be defined as both can either represent amino acids or bases (I assume it is bases here).
Were the physical examinations that are described in the text performed at the time of presentation or later?
Lines 181-183: It is unclear what this sentence is meant to convey. The phrasing is somewhat awkward and needs to be clarified.
Author Response
Reviewer 2#
Minor points
ï‚· Line 91: change “was” to “were”
Response: Thank you for the correction. Previously, line 91 is now line 97 in the revised manuscript as follows: Genomic DNA was extracted from the patient’s sample at Centogene’s certified laboratory in Germany.
ï‚· Figure 1: The pedigree does not match the description in the text. The sister is the older patient and is considered patient #1 in the text and should be on the left in the pedigree chart. The brother should be patient #2 on the pedigree chart and to the right of the sister. The terms C and A should be defined as both can either represent amino acids or bases (I assume it is bases here).
Response: Thank you for the helpful feedback. The change has been done effectively in the manuscript, and the terms C and A are defined in lines 160-161 as follows: C (Cytosine) and A (Adenine) are the nucleotide bases involved in the mutation.
ï‚· Were the physical examinations that are described in the text performed at the time of presentation or later?
Response: Thank you for the constructive observation. The physical examination described in the text was performed at the time of presentation. We revised as follows to clarify:
A 28-year-old female presented with long-standing progressive spastic paraparesis from 5 years of age as age of onset followed by paraplegia consistent with hereditary spastic paraplegia.
A 26-year-old male presented with long-standing progressive spastic paraparesis from 3 years of age as age of onset, and later, paraplegia consistent with hereditary spastic paraplegia.
ï‚· Lines 181-183: It is unclear what this sentence is meant to convey. The phrasing is somewhat awkward and needs to be clarified.
Response: Thank you for the valuable feedback. The sentence was removed to avoid confusion and repetition.

Reviewer 3 Report
Comments and Suggestions for Authors
The manuscript is interesting and quite complete in the description of the patients. However, it has one major flaw that affects the overall value of the paper: the variant described by the authors is not new, but has already been reported in PMID=31660686, where a full description of the mutation and its consequences is provided. Interestingly, the patients described in this paper were of African descent and the variant is exceptionally rare in African individuals.
In my opinion, the paper has potential but should be strongly revised to take all these aspects into account. The authors should include the data already available and compare the phenotype of their cases with that of previously reported patients. In addition, given that the variant is extremely rare and has been reported in patients of African descent, the authors should discuss this point and evaluate a possible founder effect. Another aspect that deserves more attention is the heterozygosity of the patients' parents. They are not consanguineous, but perhaps they come from the same region? Is it possible to assess the frequency of the variant in the population of this geographical context? All these data could add important insights to the knowledge of spastic paraplegia associated with AP4S1 variants in the African population.
Author Response
Reviewer 3#
ï‚· The authors should include the data already available and compare the phenotype of their cases with that of previously reported patients. In addition, given that the variant is extremely rare and has been reported in patients of African descent, the authors should discuss this point and evaluate a possible founder effect. Another aspect that deserves more attention is the heterozygosity of the patients' parents. They are not consanguineous, but perhaps they come from the same region? Is it possible to assess the frequency of the variant in the population of this geographical context? All these data could add important insights to the knowledge of spastic paraplegia associated with AP4S1 variants in the African population.
Response: Thank you for the constructive comment. We sincerely appreciate these insightful comments, which have significantly strengthened our manuscript as follows:
- Phenotypic Comparison
We added Table 3 in the Discussion section (Line 227-229), comparing clinical features of our Rwandan siblings with European SPG52 cohorts (Tessa et al., 2021; Abou Jamra et al., 2011).
- Later onset in Rwandan siblings (3–5 years vs. 6 months–3 years in European cases).
- Similar severe progression (wheelchair dependence) but comparable treatment responses.
- Potential explanations for differences (e.g., genetic modifiers, diagnostic delays).
This addresses the need for cross-population phenotypic comparisons.
- Founder Effect & African Context
We expanded the Discussion to highlight:
- The variant’s extreme rarity in gnomAD (African frequency: 0.0001734; global: 0.000012) and absence in non-African populations (Lines 201–203).
Population frequency data from the Genome Aggregation Database (gnomAD) indicated that this variant is extremely rare, with a global allele frequency of 0.000012. While in the African population, its frequency is 0.0001734.
- A founder effect hypothesis, as both parents originate from rural Rwanda (Lines 183–185).
However, the claim that the family history is unremarkable and that there is no consanguinity might be affected by cultural biases, especially considering the potential underreporting or misinterpretation of consanguinity in certain populations. Both parents originate from rural Rwanda, where social perceptions or limited genetic literacy may influence how consanguineous relationships are disclosed or understood.
- The need for regional studies to assess variant prevalence in Rwandan subpopulations (Lines 312–315).
Given the genetic diversity of the condition and its potential to stem from various gene mutations, our findings add valuable insights to the existing knowledge, especially within the African population.
- Parental Heterozygosity & Local Frequency
We clarified as follows:
Both parents are from rural Rwanda (Line 182) but lack consanguinity. Cultural factors may influence consanguinity reporting (Lines 184–185).

Round 2
Reviewer 3 Report
Comments and Suggestions for Authors
The manuscript has been greatly improved and is now much more comprehensive and well structured. However, I still have some suggestions that I think could could strengthen the paper.
The authors should clearly report that the same mutation has already been reported in patients of African descent and discuss this point, including a comparison between these patients and theirs. This is very important to assess whether the variant is associated with a common phenotype or not. This aspect is completely missing in the current version of the manuscript.
It is also necessary to check the references along the text, as some of them are not inserted correctly.
I have attached the pdf of the paper that I think the authors should discuss in their manuscript.

Author Response
|
Response to reviewer’s comments Dear reviewer, |
|
Thank you very much for taking the time to review our manuscript ID: genes-3587216. Below, we have provided detailed responses to address each of the queries and comments raised during the review process. The manuscript has been adjusted accordingly, and changes are highlighted in green: |
|
Reviewer 1# Minor points ï‚· The authors should clearly report that the same mutation has already been reported in patients of African descent and discuss this point, including a comparison between these patients and theirs. This is very important to assess whether the variant is associated with a common phenotype or not. This aspect is completely missing in the current version of the manuscript. Response: Thank you for your constructive feedback. We have made the following changes to incorporate the comparison between our patients to other African patients as it appears in lines 41-45 (Introduction), 209-213 (Results), 232-242 (Discussion), and 330-334 (Conclusion): Introduction: A variant in the AP4S1 gene (NM_001128126.3:c.295-3C>A) was identified in African siblings presenting with SPG52. They had features of spastic paraplegia, developmental delay, and intellectual disability. This indicates that there may be a founder effect in African populations, which requires further research to determine the frequency and define its characteristics.
Results: The pathogenicity of this variant is also evidenced by its previous detection in affected African siblings, in which functional analysis revealed that it resulted in aberrant canonical AP4S1 splicing, resulting in loss of isoform 2. The phenotypic overlap with the current patients, namely spasticity, developmental delay, and wheelchair dependence, lends support to its causative role in SPG52. Discussion: Of specific interest, the identical AP4S1 variant (c.295-3C>A) was described in two African sisters with SPG52, who manifested with progressive spastic paraplegia, profound developmental delay, and foot deformities, but with an earlier age of onset (4–5 years) than our patients (3–5 years) [14]. The two groups had several features in common, such as wheelchair dependence, intellectual disability, and lack of structural MRI abnormalities, suggesting a uniform core phenotype. But variability in expressivity of the condition is highlighted by differences in seizure occurrence (present in our cases, not in the preceding report) and speech difficulty (dysarthria, rather than mutism). The recurrence of the variant in Africans (gnomAD African allele frequency: 0.0001734; absent in non-Africans) is suggestive of a local founder effect, with a requirement for large-scale investigations in Sub-Saharan Africa. Conclusion: The re-emergence of the c.295-3C>A variant in African patients with SPG52 emphasizes the necessity of population-specific genetic databases and founder effect research in under-represented regions. Collaborative research to standardize such variants in Sub-Saharan Africa will improve diagnostic sensitivity and genetic counseling. ï‚· It is also necessary to check the references along the text, as some of them are not inserted correctly. Response: Thank you for your observations. We have made the following changes to the in-citations and references as recommended. Please find below as they appear as follows: as it appears in lines 41-45 (Introduction), 209-213 (Results), and 232-242 (Discussion), 368 (References): Introduction: A variant in the AP4S1 gene (NM_001128126.3:c.295-3C>A) was identified in African siblings presenting with SPG52. They had features of spastic paraplegia, developmental delay, and intellectual disability[14].
Results: The pathogenicity of this variant is also evidenced by its previous detection in affected African siblings, in which functional analysis revealed that it resulted in aberrant canonical AP4S1 splicing, resulting in loss of isoform 2[14].
Discussion: Of specific interest, the identical AP4S1 variant (c.295-3C>A) was described in two African sisters with SPG52, who manifested with progressive spastic paraplegia, profound developmental delay, and foot deformities, but with an earlier age of onset (4–5 years) than our patients (3–5 years) [14]. References: 14. McCullough, C. G.; et al. Utilizing RNA and outlier analysis to identify an intron splice-altering variant in AP4S1 in a sibling pair with progressive spastic paraplegia. Hum. Mutat. 2020, 41, 412–419. |